# The Mechanics of Bioinspired Stiff-to-Compliant Multi-Material 3D-Printed Interfaces

**DOI:** 10.3390/biomimetics7040170

**Published:** 2022-10-18

**Authors:** Dolev Frenkel, Eran Ginsbury, Mirit Sharabi

**Affiliations:** Department of Mechanical Engineering and Mechatronics, Ariel University, Ariel 407000, Israel

**Keywords:** interface, bioinspired materials, stiffness, strength, biomimetics, 3D printing, mechanical testing, stiff-to-compliant

## Abstract

Complex interfaces that involve a combination of stiff and compliant materials are widely prevalent in nature. This combination creates a superior assemblage with strength and toughness. When combining two different materials with large stiffness variations, an interfacial stress concentration is created, decreasing the structural integrity and making the structure more prone to failure. However, nature frequently combines two dissimilar materials with different properties. Additive manufacturing (AM) and 3D printing have revolutionized our engineering capabilities concerning the combination of stiff and compliant materials. The emergence of multi-material 3D-printing technologies has allowed the design of complex interfaces with combined strength and toughness, which is often challenging to achieve in homogeneous materials. Herein, we combined commercial 3D-printed stiff (PETG) and compliant (TPU) polymers using simple and bioinspired interfaces using a fused deposition modeling (FDM) printer and characterized the mechanical behaviors of the interfaces. Furthermore, we examined how the different structural parameters, such as the printing resolution (RES) and horizontal overlap distance (HOD), affect the mechanical properties. We found that the bioinspired interfaces significantly increased the strain, toughness, and tensile modulus compared with the simple interface. Furthermore, the more refined printing resolution elevated the yield stress, while the increased overlap distance mostly elevated the strain and toughness. Additionally, 3D printing allows the fabrication of other complex designs in the stiff and compliant material interface, allowing various tailor-designed and bioinspired interfaces. The importance of these bioinspired interfaces can be manifested in the biomedical and robotic fields and through interface combinations.

## 1. Introduction

The combination of dissimilar materials in a single structure is an effective approach to improve the mechanical performance. The emergence of 3D printing technologies has enabled and facilitated the design of these multi-material structures while highlighting the diverse interfacial possibilities [1,2]. These multi-material 3D printing technologies hold potential in many areas, such as in biomedical, soft electronics, optoelectronics, aerospace, and robotics applications [3,4].

Bioinspired interfaces combining stiff-to-compliant materials can improve these multi-material structures, as widely prevalent in natural materials. This combination creates a superior assemblage with strength and toughness [5,6]. For example, in the insertion site, the binding between the tendon and bone allows proper load transfer between the stiff bone (with an elastic modulus of ~20 GPa) and compliant tendon, whose elastic stiffness is 2–3 orders of magnitude lower [7,8]. Moreover, in bone and nacre, a compliant protein matrix is one of the mechanisms that allow damage tolerance and superior toughness [9,10]. In fish armor and sea sponges, interlocking mechanisms, fractal designs, and sutures enhance the interfacial stiffness and strength and increase the resistance to crack propagation [11,12,13].

When combining two different materials with large stiffness variations, an interfacial stress concentration is created, decreasing the structural integrity and making the structure more prone to failure. However, nature frequently combines two dissimilar materials with different properties [5].

Furthermore, the fracture toughness of brittle materials can be improved using interfacial engineering. i.e., the presence of soft and compliant materials in the interface between stiff constituents can significantly increase the fracture toughness through structural mechanisms such as the fractal design and crack deflection [14].

Bone–tendon and bone–ligament interfaces are responsible for attaching muscles to bones to transfer muscle forces and provide joint stability, respectively. These multi-material interfaces (entheses) carry large loads using diverse structural mechanisms. One of these mechanisms includes the overlap of the collagen fibers that intrude into the bone, providing a firmer hold between the stiff bone and compliant tendon or ligament while allowing an effective transition between materials with different mechanical properties at several length scales [7].

Biomimetics and bioinspiration have brought about numerous advances in the design of materials and interfaces. Using traditional processing techniques, mimicking the stiff-to-compliant native interfaces based on their complex structure, functional gradient, and composition is a genuine engineering challenge. However, additive manufacturing (AM) and 3D printing have revolutionized our engineering capabilities relating to the combination of stiff and compliant materials [15]. The emergence of multi-material 3D printing technologies allows the design of complex interfaces with combined strength and toughness, which is often challenging to achieve in homogeneous materials [6,16]. Although restricted to polymers, the latter is a unique approach for combining materials with elastic properties varying over three orders of magnitude. This combination demonstrated increased toughness similar to natural materials [14,16,17,18,19,20].

Different 3D printing technologies have been established over the last few years [15,21]. However, multi-material 3D printing is often achieved using state-of-the-art polyjet printers [14,18,20,22]. In this method, photocurable polymeric inks are pre-blended to fabricate polymeric materials using specific tailor-designed polymers [18].

The more common and affordable fused deposition modeling (FDM) 3D printers are not commonly used for 3D-printed multi-material interfaces.

In this study, we combined commercial 3D-printed stiff (PETG) and compliant (TPU) polymers using simple and bioinspired interfaces using an FDM printer and characterized the mechanical behavior of these interfaces. We used digital image correlation (DIC) to quantify the displacements and strains of the multi-material printed samples. Moreover, we examined how the different structural parameters, such as the printing resolution (RES) and horizontal overlap distance (HOD), affect the different mechanical properties in tailor-designed stiff–compliant 3D-printed interfaces.

## 2. Materials and Methods

### 2.1. Sample Preparation

All samples were designed using Solidworks (Dassault Systemes, France) and printed with a single printing head using an FDM Ender 5 printer (Creality 3D; Shenzhen, China). The 3D printer has a resolution of 50 μm in the z-direction (thickness) and 100 μm in the x and y-direction.

Polyethylene terephthalate glycol (PETG, eSUN, Shenzhen, China) was used for the stiff material and thermoplastic polyurethane (TPU, eSUN, Shenzhen, China), a rubber-like compliant material, was used for the compliant material. The CAD file was saved as a stereolithography (STL) file and sliced using Ultimaker Cura (Ultimaker B.V., Utrecht, The Netherlands) software to obtain the G code for the printer. The printing was performed in an enclosed chamber in a temperature-controlled environment with 100% infill.

The interface was controlled via the resolution (RES, layer thickness) and the horizontal overlap distance (HOD, Figure 1). The G-codes for layer 1 and G layer 2 (different materials) were merged into a single file to get a sequence of alternated layers (Appendix A). The overlap distance included the same amount of layers of each material, and the resolution was affected by the applied pressure by the printer (Figure 1). The 3D-printed sample dimensions were 1 mm in thickness, 5 mm in width, and 16.6 mm in gage length (Figure 2).

### 2.2. Mechanical Testing

The tensile testing was performed using the μTS load frame (Psylotech, Evanston, IL, USA) using 222 N and 1600 N load cells. The samples were stretched to failure at a rate of 3 mm∙min^−1^ under displacement control. The 1600 N load cell was used for the PETG samples and the 222 N load cell was used for the TPU and combined material specimens. The sample dimensions were measured using a digital caliper and a micrometer.

Here, 2D digital image correlation (DIC) was applied initially to examine the behavior of the combined interface. Due to the large deformations of the TPU, it was applied only to representative samples. DIC measurements and analyses were employed using 12-bit CCD digital camera (MER-503-20GM-P, Daheng, China) with a macro lens (100 mm F2.8, Nikon) and GOM Correlate software 2019 (GOM GmbH, Braunschweig, Germany). The 12-bit CCD camera consisted of a trigger that was connected to the Psylotch load frame for load data synchronization between the tensile machine and the camera. The test data were recorded using a camera rate of 15 Hz and were synchronized with the loading frame measurements. The sample preparation included spraying a very delicate thin layer of white paint and then black paint on top of the white one to achieve adequate contrast and a random speckle pattern to get a good contrast for the DIC post-processing algorithm (Figure 3).

### 2.3. Mechanical Measurements

We used engineering stresses and strains, whereby stress was defined as the force divided by the initial cross-sectional area and strain as the displacement divided by the initial gage length. The yield stress was calculated at a strain of 0.05 mm/mm. The ultimate stresses and strains were defined as the maximum stresses and strains, respectively. The moduli were calculated as the slope of the curve up to 0.025 strain. The maximal strain was taken as the failure strain. The tensile toughness was calculated as the area under the stress–strain curve up to failure using the trapezoid method.

### 2.4. Statistical Analysis

For each measurement, the mean and standard deviation were calculated. GraphPad Prism 9^®^ was used to perform a one-way ANOVA with multiple comparisons and with a Fisher LSD test analysis or unpaired *t*-tests. Statistical significance was defined as *p* < 0.05.

## 3. Results

The TPU, PETG, simple, and bioinspired interfaces were 3D-printed and tested mechanically. Three resolutions (RES = 0.10, 0.20, and 0.25 mm), three horizontal overlap distances (HOD, D = 1, 2, and 3 mm), and their combinations were printed and tested (Figure 1 and Figure 2, Table 1).

The samples with multi-material interfaces demonstrated a minor increase in thickness and width compared with single material 3D printing due to the accuracy of the printer.

Representative samples with bioinspired interfaces were captured and measured using the DIC method (Figure 3). The displacements and strains of a representative sample with a bioinspired interface were measured at t = 0, 55, and 135 s (with corresponding stresses of 0, 7.3 and 8.5 MPa, respectively) (Figure 4). As expected, the TPU was mainly deformed, while the PETG stayed stiff. The failure started at the interface, which was subjected to large deformations (up to ~25% strain). The transition between the stiff and compliant materials can be clearly seen in the DIC results (Figure 4). However, the DIC algorithm we used was limited to deformation and strain of approximately 5 mm and 25%, respectively. Above these values, the DIC algorithm could trace the speckle pattern changes.

The overall mechanical behavior results of the PETG, TPU, and combined samples with the simple and bioinspired interfaces are presented in Figure 5. The 3D-printed TPU samples demonstrated large deformations and toughness and relatively small tensile moduli. The PETG samples demonstrated a large tensile modulus and small deformations (Figure 5 and Figure 6). When combining these materials, the stress concentration in the interface and the failure started in the binding area.

The simple interface was stiffer by 230% than the TPU sample (*p* < 0.005) but less stiff than the PETG sample by an order of magnitude (*p* < 0.0001, Figure 6). The maximal strain in the simple interface was larger by 177% than the PETG.

The bioinspired interface (HOD2, RES0.2) demonstrated significantly improved mechanical properties than the simple interface; the tensile toughness and maximum strain increased by 1259% and 626%, respectively (*p* < 0.0001), the tensile modulus increased by 137% (*p* < 0.05), the UTS increased by 202%, and the strain at UTS increased by 646% (Figure 6).

For all of the tested conditions (with different RES and HOD settings), the bioinspired interfaces demonstrated improved mechanical properties compared with the simple interface. The least influence was seen on the tensile modulus. However, the toughness and strain at UTS demonstrated an increasing trend with the increase in RES for HODs of D = 2 and D = 3 mm (Figure 7).

The influence of the resolution (RES) on the mechanical properties was tested at three different resolutions, 0.1, 0.2, and 0.25 mm, as shown in Figure 8.

More specifically, in the elastic region of the stress–strain curves, there was no major difference between the different samples (Figure 7A,B). However, with larger strains, the simple interface failed (~0.1 mm/mm).

For the same printing resolution (RES), the change in HOD presented significantly increased UTS, toughness, and strain at UTS values compared with the simple interface (Figure 8). For all HODs at RES0.1, the yield stress increased significantly (*p* < 0.0005) compared with the simple interface but not for RES0.25 (Figure 7 and Figure 8E).

The tensile modulus changes did not demonstrate a specific trend and seemed similar for all tested samples at RES0.25. For RES0.2, significant changes in the tensile modulus were observed between HOD2, HOD3, and HOD1 (*p* < 0.005) and between HOD2, HOD3, and the simple interface (*p* < 0.0005, Figure 8B). The bioinspired interface significantly influenced the toughness, with an opposite trend between RES0.1 and RES0.25 (Figure 8C). For RES0.1, the strain at UTS significantly decreased with the increase in HOD (*p* < 0.005). A similar trend was seen for RES0.2 (*p* < 0.005) but not for RES0.25 (Figure 8D).

For the UTS, toughness, and strain at UTS, the effect of the resolution also caused significant changes compared with the simple interface (Figure 9). For RES0.1 (e.g., more printed layers), increased yield stress was observed compared with RES0.25 (*p* < 0.005) (Figure 9E).

A significant increase in UTS was seen for RES0.25 for HOD3 as compared with HOD1 (Figure 9A). Larger moduli were seen for the smaller RES values (RES0.1, RES0.2) (Figure 9B). For HOD2 and HOD3, increases in toughness and strain at UTS were observed when the RES was changed from 0.1 to 0.25 (fewer printed layers) (Figure 9C,D).

The resolution was more influential when the HOD was larger (3 mm). Thus, for the UTS, at D = 3 mm, RES0.25 demonstrated a significantly stronger interface than RES0.1 (*p* < 0.05).

## 4. Discussion

The interface is essential to allow a stiff-to-compliant transition with minimal stress and maximal energy absorbance and toughness. The current research investigated the effect of a bioinspired interface design, i.e., a penetrating design, in the interface between 3D-printed combined stiff (PETG) and compliant (TPU) samples. These interfaces are widely distributed in natural materials, such as in tendon-to-bone insertion sites, which combine the stiff and hard bone with the soft and stretchable ligament. In this study, we investigated how the bioinspired interface behaves mechanically compared with its constituents and how the bioinspired design principles affect the overall tensile behavior using a simple and affordable 3D FDM printer. Unlike conventional fabrication methods, 3D printing can enable the control of the interface.

The mechanical behavior of the interface resulted from the combination of the TPU and PETG materials. TPU provides large strains, ductility, and toughness but is neither strong nor stiff, while PETG is stiff and strong but not tough and ductile. For the simple interface, involving binding between the PETG and TPU in a sharp transition, there is a plane that is subjected to stress concentrations due to a sharp transition between the stiff and compliant materials. As a composite material, the simple interface specimen demonstrated increased strains compared with the PETG and an increased tensile modulus compared with the TPU. However, its failure started in the interface, demonstrating inferior UTS, strain, and toughness values. Unlike the simple interface, the bioinspired interface, which also demonstrated a combination of these two materials, exhibited significantly improved stiffness compared with the TPU, showing even larger strains and increased toughness compared with the PETG.

Thus, the bioinspired interface was far superior to the simple interface for all tested mechanical properties, including the maximal strain, tensile modulus, UTS, strain at UTS, and tensile toughness.

The DIC results demonstrated that the TPU was mainly strained and not the PETG. The bioinspired interface increased the adhesion between the materials and carried large deformations (Figure 3). The DIC algorithm could capture up to 25% strain, which was insufficient to capture the large strains that the interface carried before sample failure (Figure 5 and Figure 7).

The 3D printing technique allowed the bioinspired design of the interface between the TPU and PETG. Inspired by natural designs, in this study we tested the influence of the printing resolution, accounting for the layer thickness and the overlap distance of the 3D-printed materials, as well as their influence on the mechanical properties. We found that all tested bioinspired interfaces significantly increased the strain, toughness, and modulus compared with the simple interface. Furthermore, the more refined printing resolution elevated the yield stress, while the increased overlap distance mostly elevated the strains and toughness. For all tested designs, the mechanical properties were influenced but the ductility and plasticity were the most affected; thus, the toughness, strains, and UTS demonstrated most of the differences compared with the simple interface.

Surprisingly, the most refined resolution (RES0.1) did not always result in the most improved mechanical properties but mostly increased the yield stress (Figure 7). The larger resolutions elevated the strains, UTS, and toughness (Figure 9). Therefore, a refined resolution is not always necessary for the interface. Regarding the overlap distances (HODs) between the materials, a distance of 2 mm resulted in improved results, but an HOD of 1 mm was not enough (Figure 8). The overlapping distance strongly influenced the strain at UTS and toughness, i.e., their values increased with the increasing distance. Therefore, the printing method that included pressure-driven extrusion and an increased interfacial surface area strengthened the binding of the materials, using a toughening mechanism similar to the one observed in the tendon-to-bone enthesis. The enthesis is optimized to reduce the peak stresses and improve the failure resistance [23]. Here, the failure of the bioinspired interfaces was jagged as it is in natural materials, unlike the straight and blunt failure in the simple interface sample (Figure 2E,F).

Different additive manufacturing techniques are used for multi-material printing, mostly state-of-the-art polyjet printing with photocurable polymers. They demonstrate improved mechanical properties of bioinspired interfaces but different results than our FDM results. However, it is important to note that different materials, interfaces, and mechanical properties were investigated. For example, Roach et al. (2019) [24] developed a new novel multi-material multi-method (m4) 3D printer and combined direct ink writing (DIW) and inkjet (IJ) methods. Using that method, they demonstrated an opposite trend to our results, since the samples with overlap interfaces demonstrated inferior toughness compared to the blunt interface samples. This difference could stem from the pressure-driven extrusion during the FDM elevating the attachments between the different materials. Furthermore, Zorzetto et al. (2020) [18] investigated the mechanical properties of polyjet 3D-printed interfaces, including blurred and sharp interfaces. They found that blending stiff and compliant materials does not lead to a homogenous blend but has a biphasic nature. This method is different from FDM; therefore, the resulting interface properties are different. The printing technology, method, and materials strongly influence the interface properties. Thus, the interface optimization is case-dependent. FDM 3D printing is well-distributed and affordable. Therefore, multi-material interface 3D printing designs can affect many industries.

These multi-material interfaces are valuable in many areas, such as biomedical, soft electronics, optoelectronics, aerospace, and robotics applications [3,4]. For example, multi-material 3D printing is used for bone and soft-tissue grafts [25,26], soft actuators [27], medical prostheses [28], soft robotics [28,29], artificial muscles [30], electronics [28,29], and even 3D-printed multi-material wheels [24].

Our main focus in this paper was to investigate the effects of bioinspired interfaces on the interfaces of stiff and compliant 3D-printed materials and the parameters that influence the different mechanical properties. Additional 3D-printed bioinspired mechanisms such as graded transitions [18,22], staggered structures [14,15,20,31,32,33], and sutures [16] can be further designed to provide superior mechanical properties to the multi-material systems. Improved technologies involving affordable FDM printers would allow nature’s complex structures to be replicated in our daily life, profiting from superior mechanical properties.

## 5. Conclusions

In this study, we demonstrated the mechanical behavior of combined TPU and PETG 3D-printed samples using a simple and affordable FDM printer. The combined samples showed mechanical behavior in between their separate constituents. The interface design significantly influenced the mechanical properties, mainly the stiffness, strain, UTS, and toughness. The simple interface was significantly inferior to the bioinspired interfaces, which included horizontal overlap between the printed layers. We found that to improve the yield stress, a more refined resolution is needed. Moreover, to increase the strain and toughness, an elevated overlap distance is required.

The importance of bioinspired interface designs can be seen in the different applications in the biomedical and robotic fields and through interface combinations.

## Figures and Tables

**Figure 1 biomimetics-07-00170-f001:**
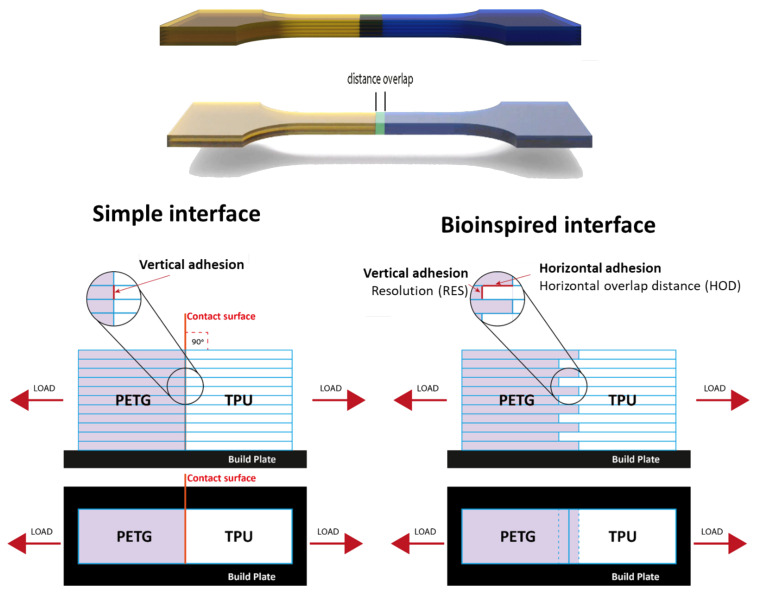
Schematic illustration of 3D-printed stiff-to-compliant interface samples. Simple interface (**left**) vs. bioinspired interface (**right**) with the effects of the resolution (RES) and horizontal overlap distance (HOD).

**Figure 2 biomimetics-07-00170-f002:**
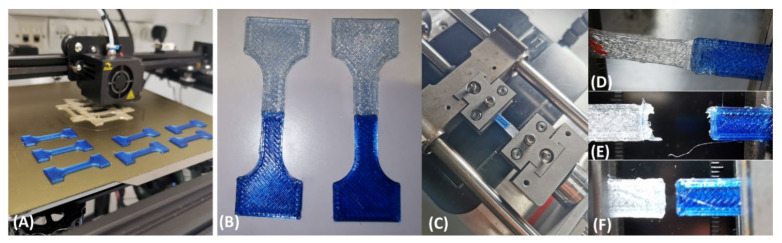
(**A**) The 3D printing of dog bones. (**B**–**F**) Combined stiff (PETG, blue) and compliant (TPU, white) samples after printing (**B**) under tensile testing and failure testing (**C**–**F**).

**Figure 3 biomimetics-07-00170-f003:**
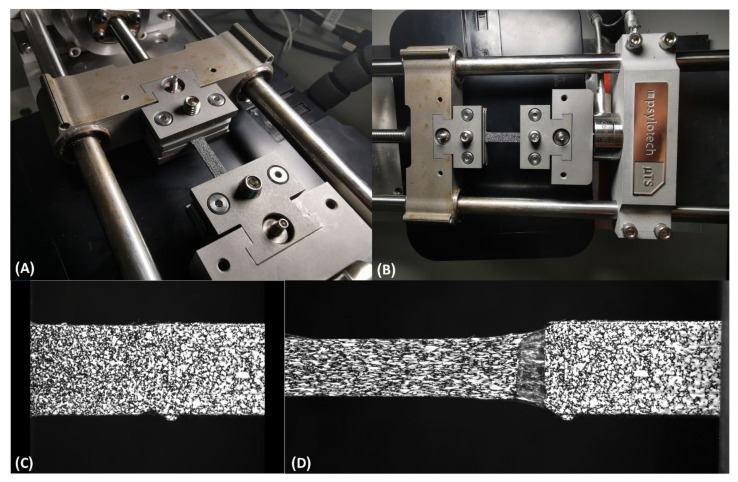
Combined stiff and compliant samples under tensile test with DIC. (**A**,**B**) Random-colored samples under tensile test for DIC post-processing. Speckle pattern of the TPU-PETG sample- before (**C**) and after (**D**) tension test.

**Figure 4 biomimetics-07-00170-f004:**
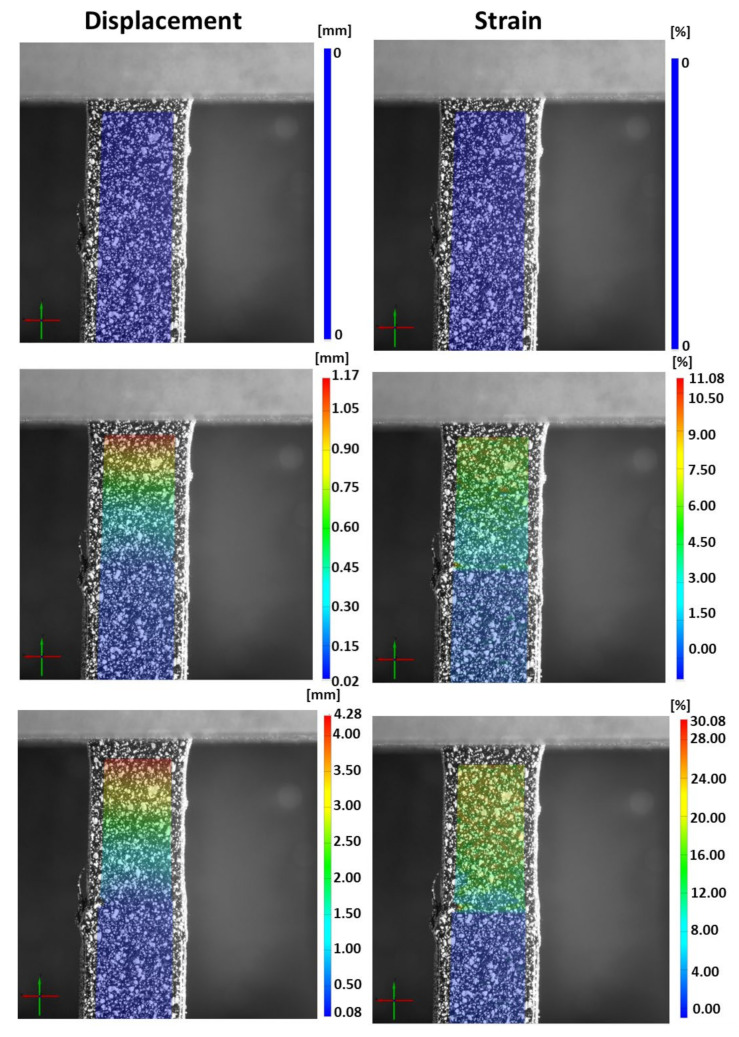
DIC measurements (strains and displacements) for of a sample with a bioinspired interface at t = 0, 55, and 135 s.

**Figure 5 biomimetics-07-00170-f005:**
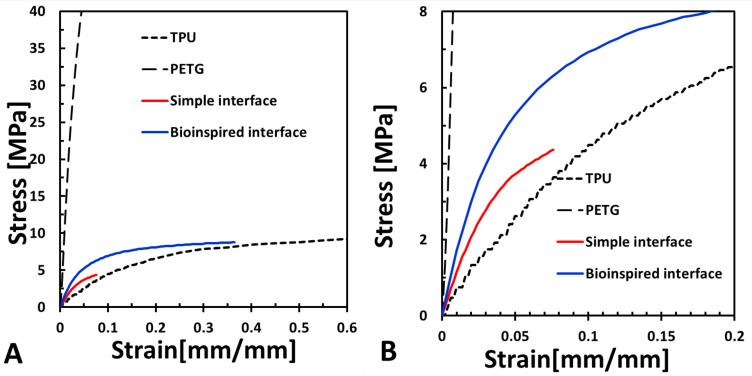
(**A**,**B**) Mechanical behavior for 3D-printed PETG, TPU, and combined samples with simple and bioinspired interfaces. (**B**) is the enlarged representation of (**A**).

**Figure 6 biomimetics-07-00170-f006:**
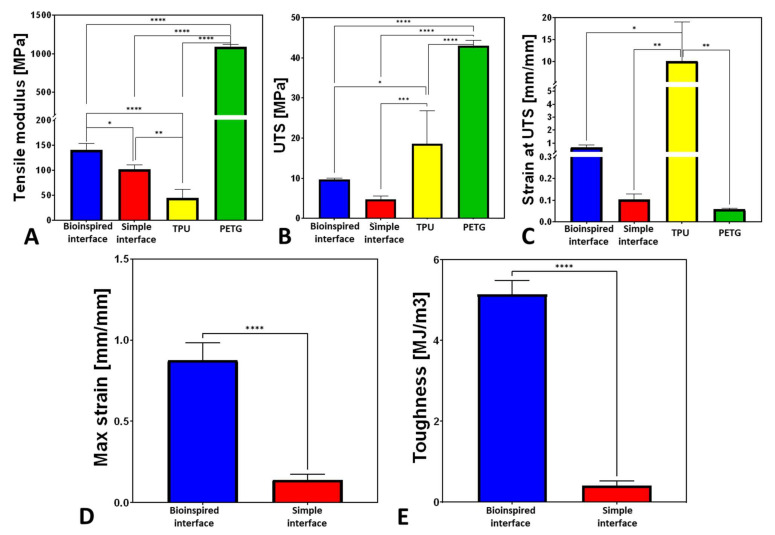
Mechanical properties of 3D-printed PETG, TPU, and combined samples with simple and bioinspired interfaces: (**A**) tensile modulus; (**B**) UTS; (**C**) strain at UTS; (**D**) maximal strain; (**E**) tensile toughness. Note: * stands for *p* < 0.05, ** for *p* < 0.005, *** *p* < 0.0005, and **** *p* < 0.0001.

**Figure 7 biomimetics-07-00170-f007:**
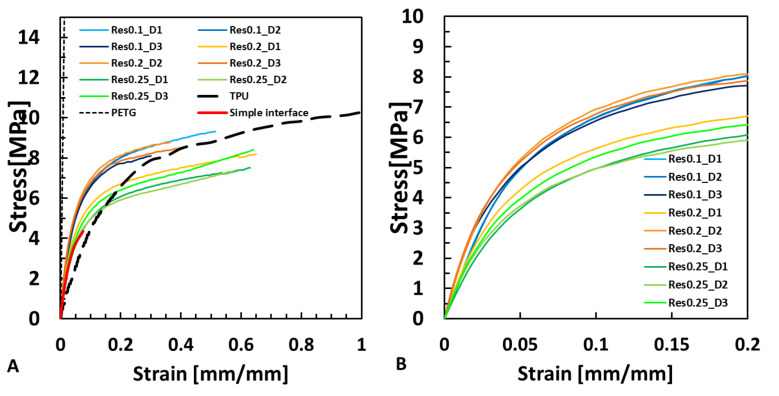
(**A**) The influence of the resolution (RES) and horizontal overlap distance (HOD, D) parameters on the mechanical behavior in comparison to the TPU, PETG and Simple interface. (**B**) The influence of RES and HOD in the mechanical behavior up to 0.2 strain.

**Figure 8 biomimetics-07-00170-f008:**
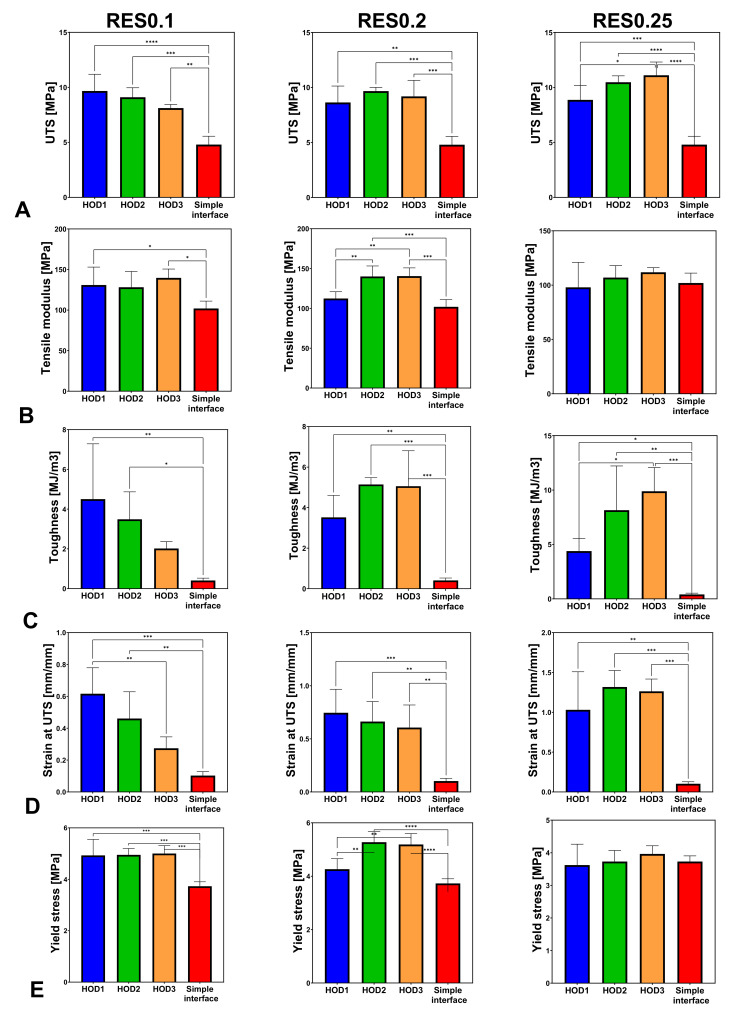
Effect of the horizontal distance (HOD) on the mechanical properties of stiff-to-compliant samples: (**A**) UTS; (**B**) tensile modulus; (**C**) tensile toughness; (**D**) strain at UTS; (**E**) yield stress (* stands for *p* < 0.05, ** for *p* < 0.005, *** *p* < 0.0005, and **** *p* < 0.0001).

**Figure 9 biomimetics-07-00170-f009:**
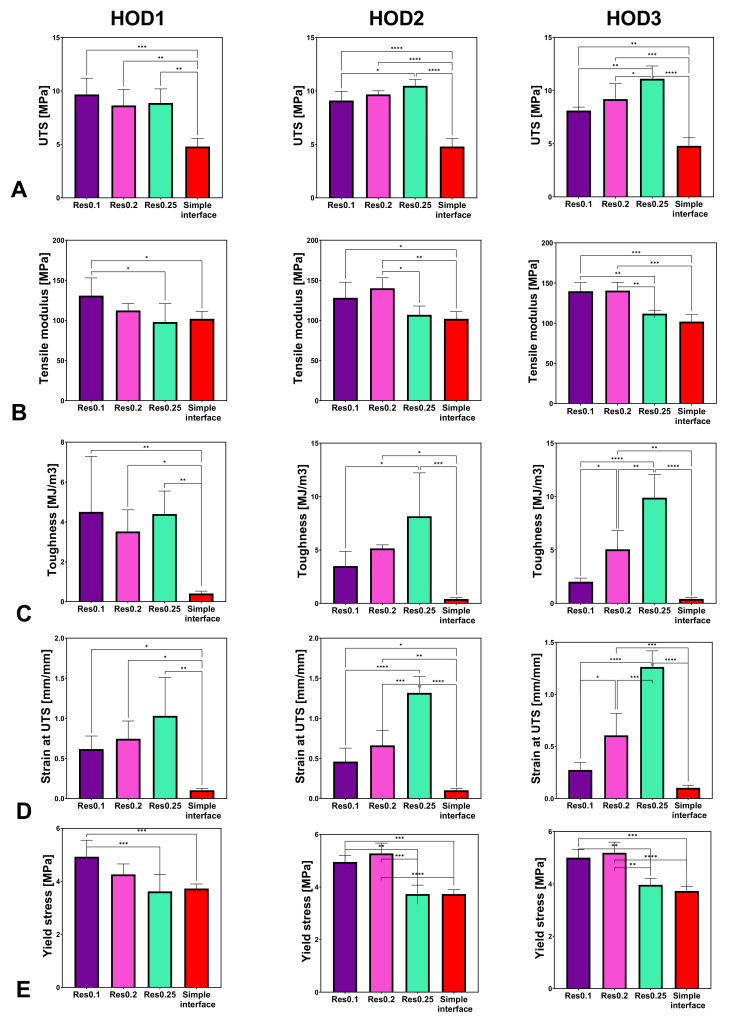
Effect of the resolution (RES) on the mechanical properties of stiff-to-compliant samples: (**A**) UTS; (**B**) tensile modulus; (**C**) tensile toughness; (**D**) strain at UTS; (**E**) yield stress (* stands for *p* < 0.05, ** for *p* < 0.005, *** *p* < 0.0005, and **** *p* < 0.0001).

**Table 1 biomimetics-07-00170-t001:** Sample description and geometry.

Sample Name	N	Thickness [mm]	Width [mm]	Gage Length [mm]	Layer Thickness [RES] [mm]	Horizontal Overlap Distance [HOD,D] [mm]
**PETG**	4	1.02 ± 0.01	4.98 ± 0.01	18.59 ± 0.86	0.2	-
**TPU**	3	1.10 ± 0.13	5.24 ± 0.36	18.59 ± 0.86	0.2	-
**Simple interface**	4	1.27 ± 0.09	5.43 ± 0.21	18.74 ± 0.38	0.2	0
**Bioinspired interface_0.1_1**	3	1.20 ± 0.12	5.26 ± 0.07	17.72 ± 1.29	0.1	1
**Bioinspired interface_0.1_2**	3	1.23 ± 0.02	5.35 ± 0.13	14.95 ± 2.30	0.1	2
**Bioinspired interface_0.1_3**	3	1.20 ± 0.01	5.22 ± 0.02	16.58 ± 0.08	0.1	3
**Bioinspired interface_0.2_1**	3	1.28 ± 0.01	5.25 ± 0.06	17.87 ± 0.34	0.2	1
**Bioinspired interface_0.2_2**	3	1.20 ± 0.06	5.24 ± 0.02	16.54 ± 0.46	0.2	2
**Bioinspired interface_0.2_3**	3	1.21 ± 0.01	5.22 ± 0.05	16.53 ± 0.58	0.2	3
**Bioinspired interface_0.25_1**	3	1.30 ± 0.09	5.25 ± 0.02	16.17 ± 2.47	0.25	1
**Bioinspired interface_0.25_2**	3	1.40 ± 0.05	5.30 ± 0.09	16.81 ± 0.69	0.25	2
**Bioinspired interface_0.25_3**	3	1.35 ± 0.05	5.27 ± 0.02	16.79 ± 0.93	0.25	3

## Data Availability

Not applicable.

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
