# Peer review of "The Mechanics of Bioinspired Stiff-to-Compliant Multi-Material 3D-Printed Interfaces"

_biomimetics, 2022, doi:10.3390/biomimetics7040170_

Round 1

Reviewer 1 Report

The authors presented an interesting dissertation on the mechanics of 3D-printed interfaces inspired by biological systems.

The paper is well written and easy-to-read. I have only a few minor questions:

1) improve the discussion on the potential applications of such interfaces, providing also practical examples

2) do the authors expect different results if using a different additive manufacturing approach? Please comment on this.

3) I'd like the authors to discuss more in detail the fracture mechanism at the interface, if possible also showing an enlargement of Figure 2 to better understand how fracture propagates

Author Response

The authors wish to thank the reviewer for their constructive review of our manuscript. We trust that the revised version of our manuscript, based on the Reviewer’s comments, is now well-suited for publication

Comment 1

“1) improve the discussion on the potential applications of such interfaces, providing also practical examples”

Response to comment 1

We added in the discussion (line 271):” These multi-material interfaces are valuable in many areas, such as biomedical, soft electronics, optoelectronics, aerospace and robotics [3,4]. For example, multi-material 3D printing of bone and soft-tissue grafts [26,27], soft actuators [28], medical prosthesis [29], soft robotics [29,30], artificial muscles [31], electronics [29,30] or even 3D printed multi-material wheels [25].”

Comment 2

“do the authors expect different results if using a different additive manufacturing approach? Please comment on this.”

Response to comment 2

We added in the discussion (line 257):” Different additive manufacturing techniques are used for multi-material printing, mostly state-of-the-art Polyjet printing of photocurable polymers. They demonstrated improved mechanical properties of bioinspired interfaces but different results than our FDM results. However, it is important to note that different materials, interfaces and mechanical properties were investigated. For example, Roach et al. (2019)[25] developed a new novel multi-material multi-method (m4) 3D printer and combined direct ink write (DIW) and inkjet (IJ) methods. Using that method, they demonstrated an opposite trend with our results since the samples with overlap interface demonstrated inferior toughness compared to the blunt interface samples. This difference can stem from the pressure-driven extrusion of the FDM method that elevates the attachment between the different materials. Furthermore, Zorzetto et al. (2020) [19] investigated the mechanical properties of Polyjet 3D printed interfaces, including blurred and sharp interfaces. They found that blending stiff and compliant materials does not lead to a homogenous blend but has a biphasic nature. This method is different from the FDM, therefore, the resulting interface properties are different. The printing technology, method, and materials strongly influence the interface properties. Thus, interface optimization is case-dependent. FDM 3D printing is well-distributed and affordable. Therefore, designing multi-material interface 3D printing can affect many industries.”

Comment 3

“ I'd like the authors to discuss more in detail the fracture mechanism at the interface, if possible also showing an enlargement of Figure 2 to better understand how fracture propagates”

Response to comment 3

Figure 2 was revised to enlarge subfigures E and F. We also added in the discussion (line 251): “The overlapping distance strongly influenced strain at UTS and toughness, i.e., their values increased with the increasing distance. Therefore, the printing method that included pressure-driven extrusion and increased interfacial surface area strengthened the binding of the materials, using a toughening mechanism similar to the one observed in the tendon-to-bone enthesis. The enthesis is optimized to reduce peak stresses and improve failure resistance [24]. Herein, the failure of the bioinspired interface is jagged as in natural materials, unlike the straight and blunt failure in the simple interface sample (Figure 2E,F).”

Reviewer 2 Report

The manuscript combined commercial 3D-printed stiff (PETG) and compliant (TPU) polymers using simple and bioinspired interfaces using an FDM printer and characterized the mechanical behavior of these interfaces. And The manuscript has examined how different structural parameters, such as the printing resolution (RES) and horizontal overlap distance (HOD), affect different mechanical properties towards tailor-designed stiff-com-pliant 3D-printed interfaces. I think this is an innovative idea, but some questions need to be improved:

(1)  This manuscript is defined as inspired by the fish armor, but the schematic diagram and even the full text do not show any structure of the fish armor, which does not allow the reader to associate the fish beetle with the prepared structure.

(2)  Please give me a clearer explanation of the relationship between bone and nacre, and how the design of this manuscript is analogous to bionics? Does the structure of the bone and nacre match the structure you printed?

(3)  Please pay attention to some writing specifications and check more confidently, such as whitespace on 152 lines and 257 lines, etc.

Author Response

The authors wish to thank the reviewer for their constructive review of our manuscript. We trust that the revised version of our manuscript, based on the Reviewer’s comments, is now well-suited for publication

Comment 1+2

 “ This manuscript is defined as inspired by the fish armor, but the schematic diagram and even the full text do not show any structure of the fish armor, which does not allow the reader to associate the fish beetle with the prepared structure.

Please give me a clearer explanation of the relationship between bone and nacre, and how the design of this manuscript is analogous to bionics? Does the structure of the bone and nacre match the structure you printed?”

Response to comments 1+2

The biomimetic interface is inspired from different natural materials, but mainly from the tendon-to-bone insertion site. Fish armor, bone and nacre are example for the use in multimaterial structures in nature and their influence on the mechanical properties. In order to highlight this issue, we thoroughly revised the introduction (line 31):  

The combination of dissimilar materials in a single structure is an effective approach for improving its mechanical performance. The emergence of 3D printing technologies enabled and facilitated the design space of these multi-material structures while highlighting diverse interfacial possibilities [1,2]. These multi-material 3D printing technologies hold potential in many areas, such as biomedical, soft electronics, optoelectronics, aerospace and robotics [3,4].

Bioinspired interfaces combining stiff-to-compliant materials can improve these multi-material structures, as widely prevalent in natural materials. This combination creates a superior assemblage between strength and toughness [5,6]. For example, in the insertion site, the binding between tendon to bone allows proper load transfer between the stiff bone (with an elastic modulus of ~20GPa) and compliant tendon whose elastic stiffness is 2-3 orders of magnitude lower [7,8]. Moreover, in bone and nacre, a compliant protein matrix is one of the mechanisms that allow damage tolerance and superior toughness [9,10]. In fish armor and sea sponges, interlocking mechanisms, fractal designs, and sutures enhance interfacial stiffness and strength and increases resistance to crack propagation [11–13].

We also added in the discussion (line 251): The overlapping distance strongly influenced strain at UTS and toughness, i.e., their values increased with the increasing distance. Therefore, the printing method that included pressure-driven extrusion and increased interfacial surface area strengthened the binding of the materials, using a toughening mechanism similar to the one observed in the tendon-to-bone enthesis. The enthesis is optimized to reduce peak stresses and improve failure resistance [24]. Herein, the failure of the bioinspired interface is jagged as in natural materials, unlike the straight and blunt failure in the simple interface sample (Figure 2E,F).

 And in line 271: “These multi-material interfaces are valuable in many areas, such as biomedical, soft electronics, optoelectronics, aerospace and robotics [3,4]. For example, multi-material 3D printing of bone and soft-tissue grafts [26,27], soft actuators [28], medical prosthesis [29], soft robotics [29,30], artificial muscles [31], electronics [29,30] or even 3D printed multi-material wheels [25].

Comment 3

(3)  Please pay attention to some writing specifications and check more confidently, such as whitespace on 152 lines and 257 lines, etc.

Response to comment 3

We revisited the paper and corrected the whitespaces.
